# Sexual and reproductive health needs of women with severe mental illness in low- and middle-income countries: A scoping review

Shilpa Sisodia[1]☯, Zara Hammond[2]☯, Jo Leonardi-Bee[1], Charlotte Hanlon[3,4], Laura Asher[1,5]*

1 Nottingham Centre for Public Health and Epidemiology, School of Medicine, University of Nottingham, Nottingham, United Kingdom, 2 Placed with NHS England Midlands Regional Public Health Directorate, Nottingham, United Kingdom, 3 Department of Psychiatry, School of Medicine, College of Health Sciences, Addis Ababa University, Addis Ababa, Ethiopia, 4 Centre for Global Mental Health, Health Service and Population Research Department, Institute of Psychiatry, Psychology and Neuroscience, King's College London, London, United Kingdom, 5 Institute of Mental Health, University of Nottingham, Nottingham, United Kingdom

☯ These authors contributed equally to this work.
* Laura.Asher@nottingham.ac.uk

**Data Availability Statement:** All relevant data are within the manuscript and its Supporting Information files.

## Abstract

### Background

This scoping review aimed to understand the extent and type of evidence in relation to sexual and reproductive health needs of women with severe mental illness (SMI) in low- and middle-income countries (LMIC) and to summarise those needs.

### Methods

Inclusion criteria were 1) focus on sexual and reproductive health needs 2) women or girls with SMI, professionals, caregivers of women with SMI and community members 3) study set in a LMIC 4) peer reviewed literature (no restriction on study date or design). Studies were identified from comprehensive searches of Medline, EMBASE, CINAHL and PsycINFO (to July 2023).

### Results

The review included 100 papers. Most studies were cross-sectional and set in hospital outpatient departments. Only 20 of 140 LMIC countries were included in this review and only 15 studies were set-in low-income countries (LIC). Included studies often had multiple focus areas and were grouped by frequency of topic into categories of HIV (prevalence, risk behaviour and knowledge), other sexually transmitted infections (STIs), sexual function, contraception use and family planning, sexual violence, fertility, pregnancy and postpartum. Included studies indicated women with SMI have worse outcomes and worse sexual and reproductive health compared to both women without SMI and men with SMI. Women with SMI were shown to have higher rates of HIV and low levels of contraception knowledge and use, with little advice offered by professionals.

**Funding:** CH and LA receive support from the National Institute for Health and Care Research (NIHR) through the NIHR Global Health Research Group on Homelessness and Mental Health in Africa (NIHR134325) and CH also receives support from the SPARK project (NIHR200842) using UK aid from the UK Government. https://www.nihr.ac.uk/ The views expressed in this publication are those of the authors and not necessarily those of the NIHR or the Department of Health and Social Care. CH also receives support from WT grants 222154/Z20/Z and 223615/Z/21/Z. https://wellcome.org/. The funders did not play any role in the study design, data collection and analysis, decision to publish, or preparation of the manuscript.

**Competing interests:** The authors have declared that no competing interests exist.

## Conclusions

This review highlights the need for a greater diversity of study methodology, robustness of ethical and consensual reporting when researching vulnerable populations and for further research on interventions and models of care aimed at addressing stigma, discrimination and improving the sexual and reproductive health of women with SMI. Future research should better represent the breadth of LMIC, investigate cultural adaptability of interventions and consider sexual health needs across the life course.

## Introduction

Good sexual and reproductive health is fundamental to a person's 'physical, mental, emotional and social wellbeing' [1–5, 84]. The United Nations Convention on the Rights of Persons with Disabilities (CRPD) sets out that access to high quality, non-discriminatory preventative and treatment services and the rights of individuals to make their own choices about their reproductive and sexual health are human rights, which should not be compromised by disability due to mental, physical, intellectual or sensory impairment [6].

Good sexual and reproductive health describes individuals achieving their full potential for pleasurable and safe sexual experiences across the life course through being empowered to maintain their own sexual and reproductive health [1, 7]. People experiencing poor sexual and reproductive health will have a greater need for health care to support them in reaching their optimal sexual and reproductive health [8]. Need for sexual and reproductive health care relates to the provision of safe, effective, accessible services, encompassing a broad range of activities including but not limited to detection, treatment and management of sexual and reproductive health-related conditions, comprehensive information and education on not only health risks but also sexual function and satisfaction, support for fertility choices including contraceptive, antenatal, childbirth, postnatal and abortion services, as well as sexual health across the life course including during and after menopause [7–9]. All aspects of sexual health needs should be experienced free from coercion, discrimination and violence [1].

High levels of unintended pregnancies, use of unsafe abortion methods, untreated sexually transmitted infections (STIs) and poor access to sexual health services are just a handful of examples illustrating that the sexual and reproductive health and healthcare needs of women living in low- and middle-income countries are not well met [8–11]. The reasons behind this are complex [10, 12]. Beliefs around sexuality, reproductive health and gender are often closely underpinned by social, cultural and political views [11, 13, 14].

Severe mental illness (SMI) refers to mental health conditions associated with substantial and enduring impacts on functioning. In this review, our focus is on psychotic disorders such as schizophrenia, bipolar disorder and severe psychotic depression. For women with severe mental illness, the compounding effect of stigma and exclusion, economic deprivation associated with mental health, gender inequality, paucity of comprehensive mental and sexual health services in LMIC and psychosocial disability provides a multiplicity of conditions which make them acutely vulnerable to poor sexual health [11, 14]. Many women with SMI are formally or informally subjected to the assumption that they lack capacity to make decisions about their own sexual and reproductive health [11, 15]. This renders women with SMI vulnerable to human right abuses such as the experience of forced or coercive reproductive interventions and being deprived of or unable to access information and services related to sexuality and reproductive health, putting them at significantly greater risk of sexual violence with often little opportunity for justice [11, 15].

This scoping review aimed to understand and quantify the extent and type of evidence available in relation to the sexual and reproductive health needs of women with SMI in LMIC. Secondary to the this, the scoping review also aimed to summarise the sexual and reproductive health needs of women with SMI in LMIC.

## Materials and methods

This scoping review was conducted using the Joanna Briggs Institute (JBI) guidance for scoping reviews [16] and reported according to the PRISMA extension for scoping reviews [17]. A pilot literature search was conducted, showing a range of literature available on the research topic and no similar review published or underway. A scoping review protocol was developed and published on Open Science Framework on 2 June 2023, DOI: 10.17605/OSF.IO/P5YA8. As this was a scoping review of published articles, ethical approval was not sought.

Inclusion criteria:

- Concept: Studies focusing on sexual and reproductive health (including but not limited to family planning, contraception, pregnancy and the postpartum period, sexual violence, sex work, STI, fertility, female orgasms, denial of sexual life), and;

- Described need (the concept of "need" referred to any aspects of need including needs that are met, unmet, quantification of need, nature of need, determinants of need and needs in relation to services). Studies focusing solely on child sexual exploitation, parenting, marriageability and marital relationships and gynaecological topics were excluded.

- Population: Studies involving women or girls (including transgender women) and non-binary people assigned female at birth with SMI (including schizophrenia spectrum disorders, bipolar disorder, psychosis and depression with psychotic features). In studies where aggregated data were presented, studies were included if over 50% of participants had SMI and if over 80% of participants were women. Studies involving families, community members and professionals that interact with the above population were also included.

- Context: Studies conducted in a LMIC as defined by the World Bank. In studies where aggregated data were presented, studies were included if over 50% of participants were from LMIC (this was a post-hoc criterion applied to one study).

- Study Design: Peer reviewed qualitative and quantitative literature were included with no restriction on study type.

Search terms were based around three elements: terms related to sexual and reproductive health, terms related to SMI and terms related to LMIC. There was no specific search term related to women to avoid excluding studies that recruited both men and women. A search for peer reviewed studies was conducted on 23rd December 2022 (re-run 7th July 2023) on EMBASE using the EBSCO platform and Medline CINAHL and PsycINFO using the Ovid platform. Due to the large amount of peer reviewed literature identified, grey literature was not searched. Studies that focused solely on child exploitation or on wider topics such as marriageability, marital relationships and gynaecological health were also excluded. There was no restriction on study date or language to ensure the review captured any changes in literature publication over time. It was decided that any titles, abstracts or full texts not available in English would be translated into English using online (DeepL or Google Translate) or native speaker translation services to allow screening. A full example of the search strategy used for the review is available in S1 File.

## Screening of studies

Studies identified from the searches were collated in Mendeley, de-duplicated and transferred into Rayyan for screening. One reviewer screened all articles with a second reviewer screening 20% of articles. Full texts of the potentially eligible studies were screened by two independent reviewers. All studies identified for screening were available with title and abstract in English language or French.

## Data extraction

A data extraction form was developed in Microsoft Excel based on the standard JBI template. Two reviewers each extracted data from 50% of the included studies. A third reviewer independently extracted data from 10% of studies for quality assurance. Data were extracted on study publication date, country, setting, study type, aim/purpose, total sample size, sample size of participants relevant to the inclusion criteria, participant characteristics, methodology, focus of study, outcome assessed and findings. Quantitative data were summarised and topic areas within each study were coded. Once coded by topic, studies were grouped by frequency of topic and categorised accordingly.

# Results

The search identified 3054 articles, which included 332 duplicates, thus resulting in 2722 articles for title and abstract screening. Full text screening was carried out for 279 articles. An article published after re-run of the database was identified and added, thus yielding 280 articles for full text screening, of which 191 studies were excluded, primarily due to not meeting the eligibility criteria (concept/described need/context/population) for inclusion or full text articles not being available.

A total of 88 studies were included from initial database searches and a further 12 papers were identified from citation searching and re-run of database searches. Therefore, the scoping review included 100 studies (Fig 1). Included studies are summarised in Table 1 and S2 File.

## Year of study

The oldest included study was from 1998 [3], whilst the most recently published were from 2023 [80, 107]. Over the 25 years that relevant studies were published, 62 were published within the last 10 years. The highest number of studies were published in 2014 (n = 10). Fig 2 displays the number of included studies by publication year.

## Context

Of the 140 LMIC globally, only 20 countries (14%) were represented in the included studies. Fig 3 illustrates the breadth of countries in the review according to the number of included studies from each of those countries. All studies focused on a single country, except for one study which included participants drawn from three South American countries [29]. Most included studies were conducted in the continents of Africa (n = 41) and Asia (n = 41), with n = 18 studies being set in the continent of South America. The Middle East was the setting for n = 22 of the included studies.

India was the country setting for the largest number of studies (n = 17). Other countries with >10 included studies were Brazil (n = 14), South Africa (n = 13) and Turkey (n = 12). This meant that 4 of the 140 (3%) LMIC globally accounted for 56% of all included studies.

The topic of studies carried out in India were heterogenous. Eight of the 14 studies were carried out in Brazil and 11 of the 13 studies carried out in South Africa focused on HIV. In

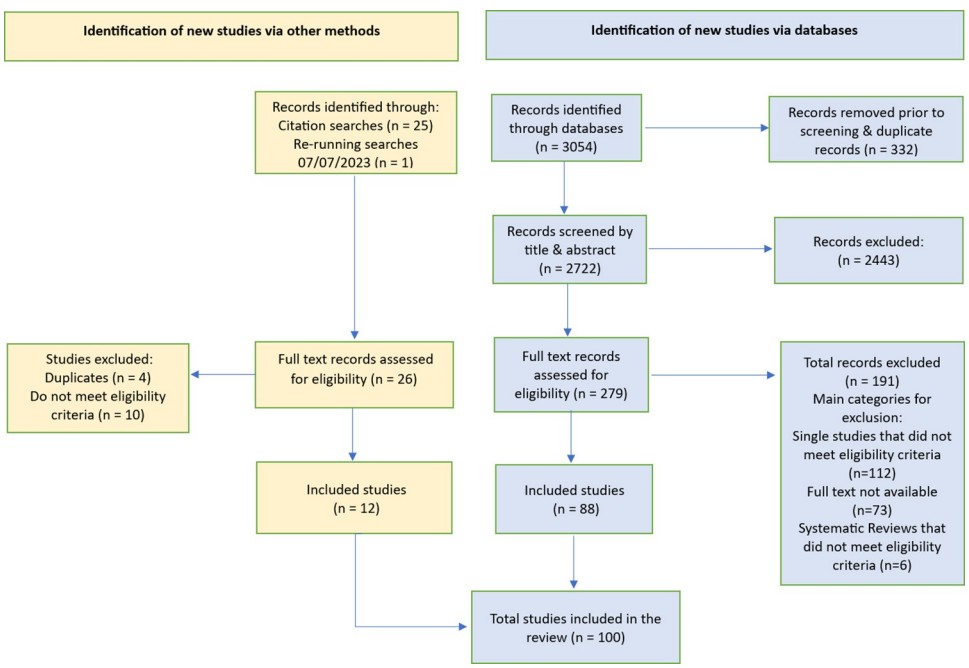

**Fig 1. PRISMA flow diagram.**

contrast, most studies set in Turkey were focused on sexuality and sexual function (n = 6) and family planning (n = 4).

The majority (n = 7) of studies in Uganda were based around HIV and HIV risk behaviours. Four of the six studies set in China were based around sexual function, three of which utilised the Arizona Sexual Experiences (ASEX) Questionnaire [59, 60, 113]. All Tunisian studies (n = 4) were focused on sexuality and all Iranian studies (n = 3) were qualitative.

Psychiatric hospital outpatient facilities were the most frequent setting for studies (n = 46), followed by psychiatric hospital inpatient settings (n = 29). In addition, a proportion of studies utilised mixed inpatient and outpatient psychiatric settings (n = 20), five of which also incorporated participants from the community. Two studies were conducted in primary care, both of which were based in China. A small number of studies were based solely in the community and not in health facilities (n = 3).

## Participants

A total of 38,510 participants were included in the studies, however some studies used the same participants so the total likely overestimates the number included. Of the total number of participants, 19,987 (52%) were relevant to the study population of this scoping review. Small numbers of men and women without SMI may be included in this total in studies that were not disaggregated by diagnosis or gender.

The most represented SMI diagnosis was schizophrenia. Twenty-eight studies focused solely on schizophrenia and people with schizophrenia were included in a further 54 studies. People with a diagnosis of bipolar disorder were included in 55 studies with an additional 8 studies exclusively focused on people with bipolar disorder.

Generally, inclusion criteria of the studies restricted participation to those 18 years of age or older. Only four studies [28, 26, 89, 102] included women aged 15-years or above and one study included those aged 16 or older [18]. The oldest recorded participant included in a study

**Table 1. Summary of included studies.**

| Author | Year | Country | Setting | Study type and methodology | Total sample size | Relevant sample size | Outcome measures |
|---|---|---|---|---|---|---|---|
| Abayomi [18] | 2013 | Nigeria | Psychiatric Hospital–inpatient and outpatient | Cross-sectional | 102 | 28 | Risky sexual behaviour (bespoke questions) |
| Abekah-Carter [19] | 2022 | Ghana | In the community | Qualitative | 20 | 20 | Sexual violence focus |
| Afe [20] | 2016 | Nigeria | Outpatient–Psychiatric Hospital | Cross-sectional | 77 | 77 | Sexual assault (bespoke questions) |
| Aboobaker [21] | 2022 | South Africa | Outpatient–Psychiatric Hospital | Cross-sectional | 368 | 181 | HIV blood test |
| Amr [22] | 2012 | Egypt | Outpatient–Psychiatric Hospital | Cross-sectional | 98 | 37 | Trauma Assessment for Adults–Brief Revised Version (TAA) |
| Aneja [23] | 2020 | India | Outpatient–Psychiatric Hospital | Case Study | 1 | 1 | n/a |
| Bagadia [24] | 2020 | India | Outpatient–Psychiatric Hospital | Qualitative (grounded theory) | 42 | 42 | Pregnancy focus |
| Ben [25] | 2013 | Tunisia | Outpatient–Psychiatric Hospital | Cross-sectional | 61 | 61 | Sexual Behaviour Questionnaire |
| Bhatia [26] | 2004 | India | Psychiatric hospital–inpatient and outpatient | Cross-sectional | 368 | 108 | Fertility (bespoke questions) |
| Bram [27] | 2014 | Tunisia | Inpatient–Psychiatric Hospital | Cross-sectional | 38 | 16 | Changes in Sexual Functioning Questionnaire |
| Bursalioglu [28] | 2013 | Turkey | Inpatient–Psychiatric Hospital | Cross-sectional | 196 | 96 | Personal Information Questionnaire (contraception use) |
| Caqueo-Urizar [29] | 2018 | Bolivia Peru Chile | Outpatient–Psychiatric Hospital | Cross-sectional | 247 | 83 | Schizophrenia Quality of Life Questionnaire (SqoL18) |
| Carey [30] | 2007 | India | Inpatient–Psychiatric Hospital | Cross-sectional | 948 | 375 | Blood test; HIV-Risk Screening Instrument (HRSI) |
| Carmo [31] | 2013 | Brazil | Psychiatric hospital–inpatient and outpatient | Cross-sectional | 2087 | 1089 | HCV seropositivity |
| Carmo [32] | 2014 | Brazil | Psychiatric hospital–inpatient and outpatient | Cross-sectional | 2206 | 1147 | HbsAg seropositivity |
| Ceylan [33] | 2019 | Turkey | Outpatient–Psychiatric Hospital Community | Cross-sectional | 186 | 186 | Attitudes, awareness and practices regarding reproductive health needs of people with schizophrenia (bespoke questions) |
| Chandra (a) [34] | 2003 | India | Inpatient–Psychiatric Hospital | Cross-sectional | 146 | 146 | Sexual Experiences Survey; HIV-Risk Screening Instrument |
| Chandra (b) [35] | 2003 | India | Inpatient–Psychiatric Hospital | Cross-Sectional | 50 | 50 | Sexual Experiences Survey; |
| Chandra (c) [36] | 2003 | India | Inpatient–Psychiatric Hospital | Cross-sectional | 618 | 429 | HIV-Risk Screening Instrument (HIS) |
| Chandra [37] | 2006 | India | Inpatient–Psychiatric Hospital | Cohort | 39 | 19 | HIV knowledge (bespoke questions) |
| Chopra [3] | 1998 | India | Inpatient–Psychiatric Hospital | Cross-sectional | 59 | 30 | HIV risk (bespoke questions); Blood test |
| Collins [38] | 2006 | South Africa | Inpatient–Psychiatric Hospital | Qualitative | 46 | 46 | Sexual violence focus |
| Collins [39] | 2009 | South Africa | Inpatient–Psychiatric Hospital | Cross-sectional | 151 | 75 | Blood test |
| Collins [40] | 2001 | South Africa | Mixed–inpatient and outpatient facilities (community and hospital) | Ethnography | 56 | 56 | Sexual violence, sexual function, contraception use focus |

(*Continued*)

**Table 1.** (Continued)

| Author | Year | Country | Setting | Study type and methodology | Total sample size | Relevant sample size | Outcome measures |
|---|---|---|---|---|---|---|---|
| Correa [41] | 2020 | Colombia | Outpatient–Psychiatric Hospital | Cross-sectional | 160 | 79 | Contraception awareness (bespoke questions) |
| De Oliveira [42] | 2012 | Brazil | Mixed–inpatient and outpatient facilities (community and hospital) | Cross-sectional | 2475 | 1277 | Sexual violence (bespoke questions) |
| Desai [43] | 2009 | India | Outpatient–psychiatric hospital | Case Study | N/a | N/a | Ethical issues in pregnant women with SMI |
| Dogu [44] | 2012 | Turkey | Outpatient–Psychiatric Hospital | Cross-sectional | 120 | 63 | Sexual function (bespoke questions) |
| Dutra [45] | 2014 | Brazil | Psychiatric Hospital–inpatient and outpatient | Cross-sectional | 2145 | 1129 | Self-reported STI |
| Eroglu [46] | 2020 | Turkey | Outpatient–psychiatric hospital | Cross-sectional | 117 | 58 | Contraception use, gravidity and parity (bespoke questions) |
| Esan [47] | 2018 | Nigeria | Outpatient–Psychiatric Hospital | Cross-sectional | 90 | 45 | Arizona Sexual Experiences Questionnaire; New Sexual Satisfaction Scale |
| Fanta [48] | 2018 | Ethiopia | Outpatient–Psychiatric Hospital | Cross-sectional | 422 | 132 | Change in Sexual Functioning Questionnaire |
| Gebeyehu [49] | 2021 | Ethiopia | Outpatient–Psychiatric Hospital | Cross-sectional | 223 | 113 | Risky sexual behaviour (bespoke questions adapted from behavioural surveillance survey) |
| Ghebrehiwet [50] | 2020 | Ethiopia | In the community | Qualitative (grounded theory) | 39 | 39 | Sexual violence focus) |
| Grover [51] | 2019 | India | Outpatient–Psychiatric Hospital | Cross-sectional | 219 | 65 | Contraception use and gravidity (bespoke questions) |
| Guimarães [52] | 2010 | Brazil | Psychiatric hospital–inpatient and outpatient | Cross-sectional | 2475 | 1277 | HIV risk behaviour (bespoke questions) |
| Guimarães [53] | 2014 | Brazil | Psychiatric hospital–inpatient and outpatient | Cross-sectional | 2237 | 1161 | HIV seroprevalence |
| Hall [54] | 2019 | Timor-Leste | In the community | Qualitative | 85 | 85 | Sexual violence focus |
| Halouani [55] | 2018 | Tunisia | Outpatient–Psychiatric Hospital | Case Control | 32 | 32 | Female Sexual Functioning Index (FSFI) |
| Hariri [56] | 2009 | Turkey | Outpatient–Psychiatric Hospital | Cross-sectional | 360 | 174 | Golombok Rust Inventory of Sexual Satisfaction |
| Henning [57] | 2012 | South Africa | Inpatient–Psychiatric Hospital | Cross-sectional | 195 | 95 | HIV rapid test; HIV ELISA test |
| Hocaoglu [58] | 2014 | Turkey | Outpatient–Psychiatric Hospital | Cross-sectional | 190 | 38 | Arizona Sexual Experiences Questionnaire |
| Hou [59] | 2016 | China | Primary Care | Cross-sectional | 607 | 279 | Arizona Sexual Experiences Questionnaire |
| Huang [60] | 2019 | China | Primary Care | Cross-sectional | 720 | 272 | Arizona Sexual Experiences Questionnaire |
| Incedere [61] | 2017 | Turkey | Outpatient–Psychiatric Hospital | Cross-sectional | 200 | 116 | Sexual violence (bespoke questions); Arizona Sexual Experiences Questionnaire |
| Joska [62] | 2014 | South Africa | Outpatient–Psychiatric Hospital | Cross-sectional | 100 | 85 | HIV appointment attendance (retrospective clinical record review) |
| Kazour [63] | 2020 | Lebanon | Inpatient–Psychiatric Hospital | Cross-sectional | 60 | 30 | Sexual Behaviour Questionnaire |
| Kesebir [64] | 2014 | Turkey | Outpatient–Psychiatric Hospital | Cross-sectional | 57 | 28 | Arizona Sexual Experiences Questionnaire; Golombok Rust Inventory of Sexual Satisfaction (GRISS) |
| Kumar [65] | 2021 | India | Outpatient–Psychiatric Hospital | Cross-sectional | 57 | 57 | Changes in Sexual Functioning Questionnaire |
| Loganathan [66] | 2022 | India | Outpatient–Psychiatric Hospital | Qualitative | 200 | 82 | Pregnancy focus |

(*Continued*)

**Table 1.** (Continued)

| Author | Year | Country | Setting | Study type and methodology | Total sample size | Relevant sample size | Outcome measures |
|---|---|---|---|---|---|---|---|
| Lundberg [67] | 2012 | Uganda | Psychiatric hospital–inpatient and outpatient | Qualitative | 20 | 13 | Sexual function and sexual violence focus |
| Lundberg [68] | 2015 | Uganda | Inpatient–Psychiatric Hospital | Cross-sectional | 602 | 343 | World Health Organization Violence Against Women Instrument; HIV blood test |
| Lundberg [69] | 2013 | Uganda | Inpatient–Psychiatric Hospital | Cross-sectional | 602 | 343 | Blood test |
| Madziro–Ruwizhu [70] | 2019 | Zimbabwe | Outpatient–Psychiatric Hospital | Cross-sectional | 270 | 142 | Blood test |
| Magalhães [71] | 2009 | Brazil | Outpatient–Psychiatric Hospital | Cross-sectional | 136 | 136 | Contraception use (bespoke questions) |
| Maling [72] | 2011 | Uganda | Inpatient–Psychiatric Hospital | Cross-sectional | 272 | 116 | Blood test |
| Mamabolo [73] | 2012 | South Africa | Inpatient–Psychiatric Hospital | Cross-sectional | 113 | 36 | Sexual violence (bespoke questions) |
| Marengo (a) [74] | 2015 | Argentina | Outpatient–Psychiatric Hospital | Cross-sectional | 126 | 63 | Contraception use (bespoke questions); London Measure of Unplanned Pregnancy (LMUP) |
| Marengo (b) [75] | 2015 | Argentina | Outpatient–Psychiatric Hospital | Cross-sectional | 126 | 63 | Contraception use (bespoke questions); HIV-risk Time Line Follow Back interview (TLFB) |
| Mashaphu [76] | 2007 | South Africa | Inpatient–Psychiatric Hospital | Cross-sectional | 63 | 13 | HIV Blood test |
| Matshoba [4] | 2021 | South Africa | Outpatient–Psychiatric Hospital | Cross-sectional | 214 | 76 | HIV knowledge and attitudes (bespoke questions) |
| Melo [5] | 2010 | Brazil | Psychiatric hospital–inpatient and outpatient | Cross-sectional | 2475 | 1277 | HIV knowledge (bespoke questions) |
| Mere [77] | 2018 | South Africa | Inpatient–Psychiatric Hospital | Cross-sectional | 201 | 121 | HIV status (retrospective clinical record review)) |
| Mirsepassi [78] | 2022 | Iran | Outpatient–psychiatric hospital | Qualitative | 21 | 4 | Sexual function focus |
| Mpango [79] | 2022 | Uganda | Outpatient–Psychiatric Hospital | Cross-sectional | 1201 | 654 | Blood Tests (HIV and Syphilis) |
| Mwelase [80] | 2023 | South Africa | Inpatient–psychiatric hospital | Cross-sectional | 294 | 92 | HIV status (retrospective clinical record review) |
| Nakhli [81] | 2014 | Tunisia | Outpatient–Psychiatric Hospital | Cross-sectional | 100 | 30 | Arizona Sexual Experiences Scale |
| Nakigudde [82] | 2013 | Uganda | Outpatient–Psychiatric Hospital Community | Qualitative | 23 | 8 | Family planning and postpartum focus |
| Negash [83] | 2019 | Uganda | Outpatient–Psychiatric Hospital | Cross-sectional | 442 | 115 | Risky sexual behaviour (bespoke questions) |
| Obo [84] | 2019 | Ethiopia | Outpatient–Psychiatric Hospital | Cross-sectional | 424 | 201 | Risky sexual behaviour (bespoke questions) |
| Olisah [85] | 2016 | Nigeria | Outpatient–Psychiatric Hospital | Cross-sectional | 255 | 133 | International Index of Erectile Function (IIEF) Questionnaire: Female Sexual Function Index (FSFI) |
| Opondo [86] | 2018 | Botswana | Inpatient–Psychiatric Hospital | Cross-sectional | 1482 | 482 | HIV status (retrospective clinical record review) |
| Ozcan [87] | 2014 | Turkey | Inpatient–Psychiatric Hospital | Cross-sectional | 292 | 292 | Contraception use and knowledge, sexual violence (bespoke questions) |

(*Continued*)

**Table 1.** (Continued)

| Author | Year | Country | Setting | Study type and methodology | Total sample size | Relevant sample size | Outcome measures |
|---|---|---|---|---|---|---|---|
| Ozcan [88] | 2018 | Turkey | Inpatient–Psychiatric Hospital Outpatient–Psychiatric Hospital Community | Cross-sectional and comparative descriptive | 349 | 149 | Prenatal Attachment Inventory; Maternal Attachment Scale |
| Pehlivanoglu [89] | 2007 | Turkey | Outpatient–psychiatric hospital | Cross-sectional | 200 | 100 | Contraception use and knowledge (bespoke questions) |
| Peixoto [2] | 2014 | Brazil | Psychiatric Hospital–Inpatient and outpatient | Cross-sectional | 1475 | 791 | Contraception use, sexual violence, HIV knowledge (bespoke questions, clinical data) |
| Pinto [90] | 2007 | Brazil | Inpatient–Psychiatric Hospital | Qualitative–Ethnography | 88 | 30 | HIV risk behaviour focus |
| Poreddi [91] | 2021 | India | Inpatient–Psychiatric Hospital | Qualitative | 20 | 20 | Sexual violence focus |
| Rezaie [92] | 2020 | Iran | Inpatient–Psychiatric Hospital | Qualitative | 42 | 42 | Pregnancy and sexual function focus |
| Sethuraman [93] | 2019 | India | Outpatient–psychiatric hospital | Cross-sectional | 96 | 96 | Modified National Family Health Survey-3 |
| Shahvari [94] | 2020 | Iran | Inpatient–Psychiatric Hospital | Qualitative | 21 | 21 | HIV risk behaviour and contraception use focus |
| Sibanyoni [95] | 2022 | South Africa | Psychiatric hospital–inpatient and outpatient | Qualitative | 23 | 23 | Pregnancy and family planning focus |
| Simiyon [96] | 2016 | India | Outpatient–Psychiatric Hospital | Cross-sectional | 63 | 63 | Marital Quality Scale Female Sexual Function Index (FSFI). |
| Singh [97] | 2009 | South Africa | Inpatient–Psychiatric Hospital | Cross-sectional | 206 | 88 | HIV Blood tests–ELISA |
| Souaiby [98] | 2020 | Lebanon | Outpatient–Psychiatric Hospital | Cross-sectional | 95 | 13 | Psychotropic-Related Sexual Dysfunction Questionnaire (PRSexDQ) |
| Souto [99] | 2011 | Brazil | Inpatient–Psychiatric hospital Outpatient–Psychiatric Hospital | Cross-sectional | 2380 | 1232 | Self-reported HIV testing |
| Terzian [100] | 2006 | Brazil | Outpatient–Psychiatric Hospital | Cross-sectional | 167 | 65 | Fertility rate |
| Tharoor [101] | 2015 | India | Outpatient–Psychiatric Hospital | Cross-sectional | 136 | 67 | Psychotropic Related Sexual Dysfunction Questionnaire (PRSexDQ-Salsex) |
| Tomruk [102] | 2006 | Turkey | Outpatient–Psychiatric Hospital | Cross-sectional | 200 | 100 | HIV risk behaviour (bespoke questions) |
| Tumwakire [103] | 2022 | Uganda | Inpatient–Psychiatric Hospital | Qualitative | 14 | 14 | Sexual function, contraception use and sexual violence focus |
| Tunde [104] | 2013 | Nigeria | Outpatient–Psychiatric Hospital | Cross-sectional | 100 | 100 | Family planning experience/knowledge (bespoke questions) |
| Wainberg [105] | 2007 | Brazil | Psychiatric hospital–inpatient and outpatient | Qualitative | 88 | 30 | Sexual function and contraception use focus |
| Wainberg [106] | 2008 | Brazil | Outpatient–Psychiatric Hospital | Cross-sectional | 98 | 50 | Sexual Risk Behaviour Assessment Schedule (SERBAS) |
| Yang [107] | 2023 | China | Inpatient–Psychiatric Hospital Outpatient–Psychiatric Hospital | Qualitative–phenomonological | 20 | 10 | Sexual function focus |
| Yu [108] | 2022 | China | Psychiatric Hospital–inpatient and outpatient | Qualitative–phenomonological | 15 | 15 | Fertility, pregnancy, post-partum focus |
| Zerihun [109] | 2020 | Ethiopia | Outpatient–General Hospital (psych dept) | Cross-sectional | 422 | 422 | Gravidity, contraception use and knowledge (bespoke questions) |

(*Continued*)

**Table 1.** (Continued)

| Author | Year | Country | Setting | Study type and methodology | Total sample size | Relevant sample size | Outcome measures |
|---|---|---|---|---|---|---|---|
| Zerihun(a) [110] | 2021 | Ethiopia | Outpatient—Psychiatric Hospital | Nested qualitative | 16 | 16 | Pregnancy, family planning, sexual violence focus |
| Zerihun(b) [111] | 2021 | Ethiopia | Outpatient—General Hospital (psych dept) | Cross-sectional | 422 | 422 | Adapted WHO Intimate Partner Violence tool |
| Zhang (a) [112] | 2018 | China | Inpatient—Psychiatric Hospital | Cross-sectional | 87 | 33 | Syphilis seroprevalence |
| Zhang (b) [113] | 2018 | China | Inpatient—Psychiatric Hospital | Cross-sectional | 118 | 63 | Arizona Sexual Experience Scale |

was 83 years. Reporting of age ranges within studies was variable and some studies reported mean age of participants only. Where age range was reported, it was noted that only a handful of studies included women of menopausal age and older.

Participants without lived experience of SMI were included in 11 studies [19, 33, 38, 40, 50, 54, 78, 82, 94, 103, 105]. These participants included psychiatric nurses, midwives, administrators of mental health care facilities, caregivers or family members of people with SMI, government decision makers, mental healthcare providers, and people living in communities that include people with SMI.

## Ethics

Included studies were searched for evidence of ethics approval, informed consent and assessment of capacity to consent. Nineteen of the 100 included studies did not report whether ethical approval had been obtained [19, 23, 25, 34–37, 43, 44, 51, 55, 65, 66, 71, 81, 90, 102, 104]. Seventy-six studies explicitly stated that informed consent had been sought from participants but of those, only 16 studies [2, 5, 31, 32, 36, 37, 42, 52, 57, 60, 67, 70, 99, 110, 111, 113]

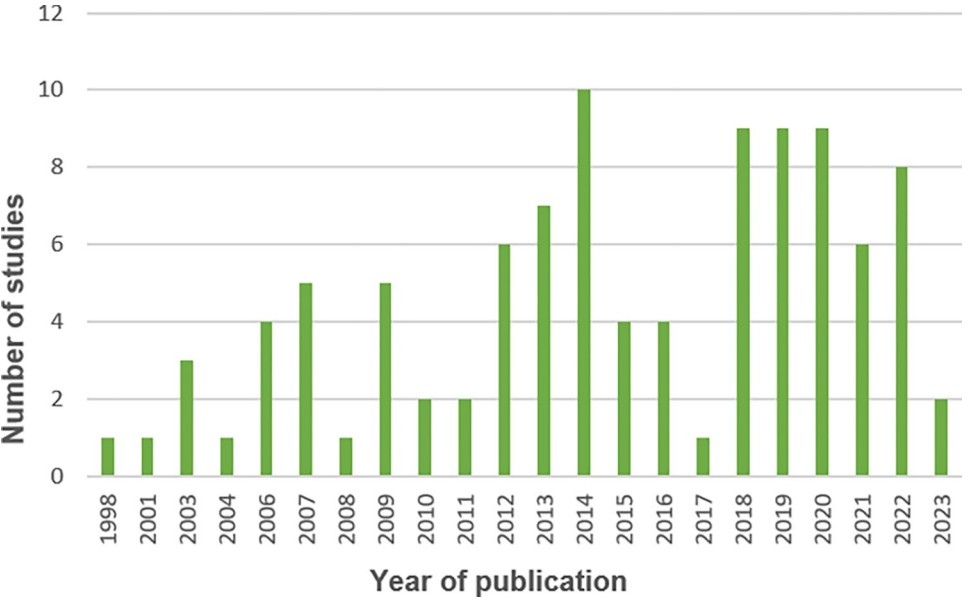

**Fig 2. Number of included studies by publication year.**

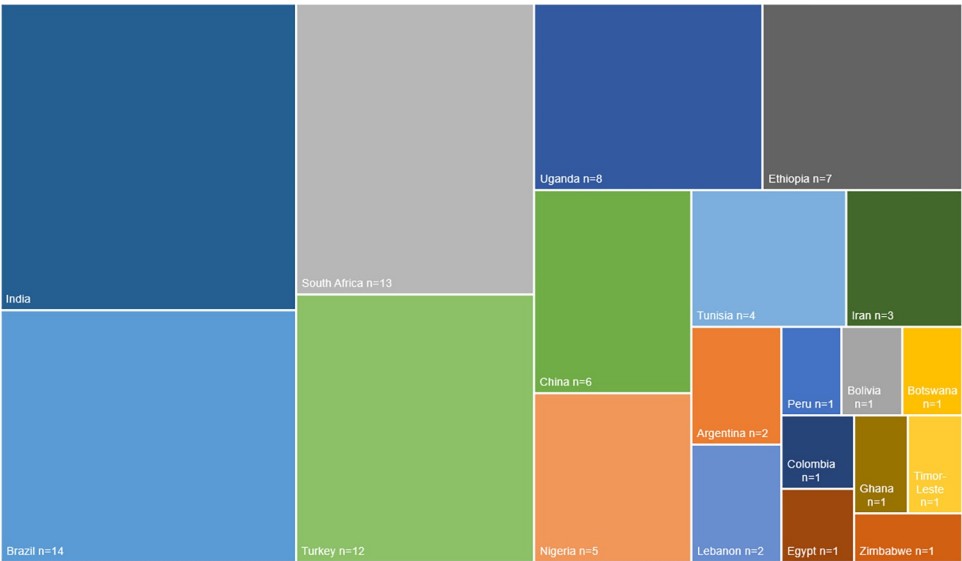

**Fig 3. Number of included studies by country.**

included a statement that capacity to consent had been assessed prior to participants being included in studies. This means that almost a quarter of studies did not provide information in their published papers as to whether consent had been obtained and most studies in the review (n = 79) did not include information as to whether capacity to consent had been assessed. A small number of papers (n = 5) did not require assessment of capacity to consent due to the nature of the study or the focus of the study being participants without lived experience of SMI [19, 33, 38, 40, 112].

Of the studies that included participants aged 15 or 16 years and older (n = 5), none explicitly stated that capacity to consent had been assessed [18, 28, 89, 97, 102] and one study did not document that consent had been obtained. All five studies did provide information indicating that ethics approval had been obtained.

Analysis by country of publication indicated that the greatest number of studies not reporting ethics approval were published in India (n = 10) [3, 23, 34–37, 43, 51, 65, 66] but Tunisia had the greatest proportion with three [25, 55, 81] of the four studies not reporting this. Of the eleven studies that did not report all three areas of ethics approval, informed consent and capacity to consent, n = 4 were published in India [23, 43, 51, 66], n = 3 were published in Tunisia [25, 55, 81], n = 2 were published in Brazil [71, 90] and n = 1 were each published in Nigeria [104] and Turkey [44]. Analysis did not detect any patterns by year of publication with regards to ethics approval, informed consent and capacity to consent assessment among the included studies.

## Methodology

Most studies (n = 76) implemented a cross-sectional design. Other quantitative study designs were one case control study [55] and one cohort study [37]. Questionnaires were the main data collection tool used in these studies. A sizeable number of studies also used blood samples as a data source.

There were 22 qualitative studies included in the review, two of which used ethnographic methodologies [40, 90]. There were two case studies in the review [23, 43]. Most of the

qualitative studies used semi-structured interviews (n = 18). The range of sample sizes for the qualitative studies was 14–200 participants.

## Study focus

Included studies frequently had multiple areas of focus. Focus areas were grouped into 6 overarching categories. The largest domains were HIV prevalence, risk behaviour and knowledge (n = 32), sexual function (n = 30), contraception use and family planning (n = 25), sexual violence (n = 21), fertility, pregnancy and the post-partum period (n = 19 studies) and non-HIV STIs (n = 11). The most relevant results from each study have been reported in S2 File.

## HIV (prevalence, risk-taking behaviour and knowledge)

All of the 14 studies that ascertained the prevalence of HIV reported it to be higher in women with SMI than in men with SMI [21, 30, 39, 53, 57, 69, 70, 72, 76, 77, 80, 86, 97]; however, not all of these studies were probabilistically sampled. Sixteen studies researched HIV risk behaviour [2, 3, 18, 34, 36, 40, 49, 53, 68, 75, 83, 90, 94, 102, 106]. Results were varied, although the majority reported that HIV risk behaviour was generally higher among women with SMI than men with SMI. Key findings from three qualitative studies [40, 90, 94] found mental healthcare providers 1.) were more concerned about the avoidance of pregnancy than the avoidance of HIV, 2.) perceived patients to be at high risk of HIV, and 3.) noted that patients may not be able to 'resist' high risk sexual behaviour and may also lack the agency to refuse unprotected sexual intercourse.

Six studies assessed the knowledge of people with SMI about HIV/AIDS [2–5, 37, 102] with inconsistent findings between the studies. Three of the studies reported that HIV knowledge in women with SMI was lower than a comparator group of men with SMI [3, 37] and a comparator group of women without SMI [102]. Women with SMI had a higher knowledge score in one study [5]. Two studies reported no significant difference in HIV knowledge score between men and women with SMI [2, 4]. Limited evidence from one study reported that a higher proportion of women with SMI attended HIV appointments compared to men with SMI [62]. A study looking at whether men and women with SMI had ever had testing for HIV reported no significant difference [99]. One qualitative study from South Africa [41] identified barriers and challenges in delivering effective HIV counselling. Examples of how the health, social and wellbeing needs of women could be better met included improved data collection on HIV prevalence among psychiatric inpatients, capacity to support health literacy and ensure test results and diagnosis are fully understood and a need to improve follow up in the community after hospital discharge, particularly where a patient returns to a rural area [41].

## STIs (not including HIV)

Most studies focused on non-HIV STIs used men with SMI as a comparator, with only one study comparing to women without SMI [75]. The prevalence of a range of STIs were reported. One study showed a similar prevalence of chlamydia between men and women [30]. In the three studies that reported the prevalence of syphilis, one showed no difference [79] whilst the other two showed prevalence to be higher in men [30, 112]. A qualitative study noted the perception of mental healthcare providers that rates of syphilis were higher in their hospital population compared to the general population [40]. Two studies reported hepatitis B prevalence, both showed a higher prevalence of hepatitis B in men with SMI than women with SMI. Similarly, a study of hepatitis C prevalence showed a higher prevalence in men [30, 32]. Prevalence of history of STI was reported by five studies [18, 32, 42, 45, 75]. Generally, studies showed a similar prevalence for history of STI between women and men but a study that compared

women with SMI to women without SMI showed a higher prevalence of STI in women with SMI [75].

The two qualitative studies included utilised healthcare provider staff participants only or in combination with people with SMI [40, 90]. One qualitative study drawing on interviews with mental healthcare providers in Brazil indicated that SMI patients' decision making in relation to risk and sexual behaviour was impaired by their mental health illness, which led to unprotected sex and in turn a higher risk of STI than the general population [90]. Dialogue with patient focus groups echoed this, indicating there may be greater sexual health needs for this population in terms of risk perception, decision making and acquisition of STIs [90].

## Sexuality and sexual function

A total of 30 studies focused on sexuality and sexual function. The most used measure of sexual function was the Arizona Sexual Experiences Scale (ASEX) which was used in eight of the included studies [47, 58–61, 64, 81, 113]. Other measures of sexual function included The Changes in Sexual Functioning Questionnaire (CSFQ) [27, 48, 65] and Female Sexual Functioning Index (FSFI) [55, 85, 96], the Sexual Behaviour Questionnaire (SBQ) [25], the Golombok Rust Inventory of Sexual Satisfaction (GRISS) [56, 64], the Psychotropic-Related Sexual Dysfunction Questionnaire (PRSexDQ) [98, 101], the Sexual Risk Behaviour Assessment Schedule (SERBAS) [106], the Schizophrenia Quality of Life Questionnaire (SQoL18) questionnaire, specifically the "sentimental life score" component (measuring satisfaction with love life) [29]. One study used a bespoke scale and found more women with SMI than men with SMI reported a problem in their sexual life [44].

Of the 18 studies that used standardised measures of sexual dysfunction and compared women with SMI to men with SMI, ten studies reported higher levels of sexual dysfunction in women with SMI than men with SMI [25, 47, 59–61, 81, 85, 98, 106, 107]. Five studies reported no significant difference between women with SMI and men with SMI [27, 48, 56, 58, 63]. Only one study reported higher levels of sexual dysfunction in men with SMI [101] and another reported lower levels of sexual satisfaction in men with SMI [29]. Limited evidence from one study indicated that self-reported levels of sexual dysfunction varied by gender when type of medication was taken into consideration [64].

Of those studies that did not use a male comparator, one study comparing groups of women with SMI taking risperidone or olanzapine medication for SMI found both groups to have high levels of sexual dysfunction [65], another study reported women with SMI to have higher levels of sexual dysfunction compared to women without SMI [55]. A third study reported 70% of women with SMI had high levels of sexual dysfunction [96] although no comparator was used.

Eight qualitative studies relating to sexuality were identified [40, 67, 78, 90, 92, 103, 105, 107]. Stigmatisation was consistently reported by women with SMI and that this was at times perpetuated by healthcare providers. Examples given were perceptions that women with SMI lacked sexual needs and the promotion of lifelong abstinence. Studies including healthcare providers reported that staff caring for women with SMI agreed they have the same sexual desires as women without SMI. Both healthcare providers and people with SMI felt that sexuality could be affected by mental illness, with both increases and decreases in sexual activity described. Women with SMI described positive experiences of sexuality to be beneficial to their mental health but also negative sexual experiences to be a result of their illness. There was limited research indicating women with SMI were keen to learn more about the interaction between SMI and their sexual fulfilment. No studies reported sexuality or sexual function needs and experiences of lesbian, gay, bisexual, transgender, intersex, queer, asexual or other sexually or gender diverse LGBTQIA+ women with SMI.

## Contraception and family planning

Twenty-five studies were based around contraception and family planning. The majority of these (n = 13) focused on contraception use [2, 28, 40, 43, 51, 53, 71, 74, 75, 90, 94, 105, 106]. Four studies, which did not include a comparison, reported large proportions of women with SMI had never used contraception, had not used contraception during their most recent sexual experiences and had no intention of using contraception during sexual intercourse [71, 87, 93, 109].

Of those studies that included a comparator, most studies used a comparator group of women with SMI rather than men with SMI. Contraception use compared to women without SMI was mixed with two studies stating no difference in contraceptive use [74, 75], two studies stating women with SMI were more likely to use contraception [28, 46] (in particular 'traditional' methods such as withdrawal and calendar methods) and one stating that women with SMI were less likely to use contraception [89]. There were similar results for studies comparing women with SMI to men with SMI; two studies found contraception use to be lower in women [2, 51] whilst one study found it to be lower in men [41].

Four qualitative studies were also identified on the topic of contraception [94, 103, 105, 106]. Healthcare professionals perceived that women with SMI were commonly engaged in unprotected sex and therefore prescribed reversible contraceptive methods for them. Women with SMI felt they may be perceived negatively by partners for using condoms. When asked about knowledge of different types of contraception, coitus interruptus, condoms and intra-uterine devices (IUDs) were generally the most cited.

A small number of studies looked at informed choices and knowledge when making contraception decisions. One study assessed whether there were differences in participants being informed about contraception based on gender and diagnosis of either schizophrenia or bipolar disorder [41]. A further study looked at the proportion of women with SMI referred to a perinatal psychiatric clinic and reported that health care providers had discussed contraception issues with them [43]. A qualitative study in Uganda looked at the autonomy of women with SMI in making contraception choices and noted that this was often absent, with consent taken from family members to force a woman to use contraception if she refused [103].

Seven studies focused on family planning advice from professionals [33, 41, 82, 87, 93, 110, 104]. The focus of included studies was varied. Most studies looked at the quantity of provision or receipt of family planning advice from the perspective of either healthcare professionals or patients. None of the included studies assessed the quality of family planning advice provided. One study included a comparison and found more women with SMI received family planning advice compared to men with SMI [41]. In one study from Turkey, 73.7% of nurses stated that patients should receive family planning education however only 23.5% stated giving family planning advice to patients [33]. In another, 21.9% of people with SMI stated they received family planning advice from professionals [87]. Two further studies concluded there was a lack of family planning advice being provided to patients from healthcare professionals [93, 104]. In a qualitative study, caregivers and postpartum women with SMI felt that family planning education should be added to an existing psychoeducation programme [82]. Two studies reported that women with SMI felt that they would prefer to receive information about family planning from mental healthcare providers [104, 110] as they are considered knowledgeable about the impact of their health conditions on family planning decision making.

Two studies (from India and Turkey) provided evidence of healthcare professionals making family planning decisions for patients [24, 33], with forced abortion being considered necessary by over 39% of health professionals working with people who have schizophrenia in one study [33]. Use of sterilisation and abortion for failed contraception was reported in a

qualitative study carried out in South Africa, which looked at the perceptions and experiences of mental health care staff [40]. Another study found that family planning considerations affected desire to take medication and were influenced by medication, with some women with SMI not wanting to take sodium valproate due to teratogenicity risk [95].

## Sexual violence

Six studies [2, 22, 42, 61, 68, 73] compared experiences of sexual violence in women with SMI to men with SMI or women without SMI. Most studies reported lifetime experience of sexual violence. Two studies reported experience of sexual violence from a partner or non-partner [68, 87]. All but one study found women with SMI to be more likely to have experienced sexual violence. An Egyptian study reported men to be more likely to report sexual violence [22]. Six studies reported the prevalence of sexual violence in populations of women with SMI, and this ranged from 10%-100% [20, 34, 35, 87, 102, 111].

There were nine qualitative studies [19, 38, 40, 50, 54, 67, 91, 103, 110] focusing on the theme of sexual violence. Two thirds of included studies utilised non-patient participants either wholly or in combination with people who have lived experience. Only three of the nine qualitative studies included exclusively people with lived experience of SMI [67, 91, 110].

All nine studies reported the vulnerability of women with SMI to sexual violence. This included women being raped by male patients, whilst inpatients in psychiatric facilities. In addition to reports of sexual violence, participants reported difficulties accessing medical care after rape for homeless women; rape resulting in unintended pregnancies and delayed discharge from medical care facilities in women who experienced sexual violence at home.

## Fertility, pregnancy and postpartum

Five studies were based around fertility [26, 51, 74, 100, 108]. These were varied in terms of the outcome being studied. Three studies looked at conception and childlessness and found similar rates between women with SMI and men with SMI or women without SMI [26, 51, 74]. A study carried out in Brazil found lower fertility rates and fecundity in women with SMI compared to women in the general population [100]. A qualitative study described difficulties in the fertility process for women with SMI, alongside parenting being described as a rewarding process that aids recovery [108].

Fourteen studies focused on pregnancy [23, 24, 41, 43, 46, 66, 87, 88, 91, 92, 95, 108–110]. Notably, the only two case studies included in this review were based around decision-making during pregnancy in the context of medication and abortion [23, 43]. Seven studies mentioned the impact of medication on pregnancy and/or the effect of withholding medication during pregnancy. One qualitative study mentioned a higher risk perception related to impacts of medication on a foetus compared to risk of unmedicated mental illness during pregnancy [110]. One Turkish study included a comparator and found that women with SMI were less likely to have experienced antenatal care during pregnancy, had a greater frequency of Caesarean birth and were subject to more birth related trauma [88]. Two studies reported that high numbers of women with SMI experience unplanned pregnancy (51.4% of women in one study [87] and 87.8% of women who had experienced pregnancy in another [109]. A study that compared women with schizophrenia to bipolar disorder [41] found those with bipolar disorder to have a lower age of first pregnancy (22 years for bipolar and 32 years for schizophrenia). Two studies reported pregnancy being discouraged for women with SMI, with 69% of Ethiopian community members felt that women with SMI should not give birth in one study [110], whilst in a study from India, there were reports of women being forced to abort or coerced not to have children [66]. A comment in another study from India reported instances of pregnancy

loss due to domestic violence [91]. One study found no significant difference in the number of pregnancies or number of lost or interrupted pregnancies between women with bipolar disorder compared to controls but that a higher proportion of women with bipolar disorder had unplanned pregnancies [46].

Three studies were based in the postpartum period [82, 88, 108]. One study compared women with SMI to women without SMI and found that women with SMI were less likely to breastfeed, more likely to express concerns about infant care, experience childbirth related trauma and rely on others to care for their babies [88]. Two studies mentioned breastfeeding: a Ugandan study discussed instances of women with SMI being told not to breastfeed as mental illness could be passed onto an infant through breastmilk [82] and some women with schizophrenia in a Chinese study stated that breastfeeding could lead to a loss of routine for women with SMI that could lead to an exacerbation of illness [108].

## Discussion

This review identified a substantial body of evidence relating to the sexual and reproductive health needs of women with SMI in LMIC. The most studied domains of sexual and reproductive health were HIV and sexual function, followed by contraception and family planning, sexual violence, fertility, pregnancy and post-partum and finally non-HIV STIs. Age ranges were not reported for all studies but where age range was known, the literature indicated a focus on adult women of reproductive age. There were no studies specifically looking at SRH needs of women with SMI during menopause. This could however be influenced by some of the studies being conducted in countries where average life expectancy for women is close to expected age for menopause [114] and that menopause was not an included term in the search strategy. SRH in girls and the SRH needs of LGBTQIA+ women with SMI were also areas with no studies identified by this review.

The review frequently found themes of women with SMI in LMIC having worse sexual and reproductive health compared to both women without SMI and men with SMI. The increasing number of studies over time may indicate this topic is receiving relatively more attention. However, only a small number of LMICs (14%) were represented in this review, signifying that the sexual and reproductive health needs of women with SMI remain underexplored in the majority of LMIC. Study focus varied by world region in this review with Middle Eastern studies mainly focused on sexuality and almost all South African studies focused on HIV. Schizophrenia was the most commonly represented SMI and the most common study type was cross-sectional (n = 76), followed by qualitative interview studies (n = 18).

A range of ethical concerns related to the sexual and reproductive health of women with SMI in LMIC were identified. In studies with mental healthcare providers, there were reports of women being forced to take medication and to undergo abortion and sterilisation. A 2019 Turkish study [33] reported that 79% of healthcare providers felt that forced abortion and sterilisation were necessary. A lack of agency was a theme throughout studies, with women reporting being pressured to not have children or breastfeed, for fear their illness would pass to their offspring. Women were also pressured by partners to avoid contraception. In addition, ethical approval, informed consent and capacity assessment procedures were not documented for a substantial minority of studies. These findings show that some women with SMI may not have their autonomy upheld and may be vulnerable to harm from family members, partners, healthcare professionals and even researchers.

Availability of family planning support and contraception seemed to be mixed. Some women with SMI may lack capacity to make decisions, such as those related to contraception. Contraception was often given without their consent in these instances. However, one study

mentioned that planning for management of relapse of mental illness when a woman was well would allow them to make their own decisions about family planning, reflecting the principles of the CRPD in terms of making decisions according to the best knowledge of an individuals' preferences [92].

## Comparison to previous literature

Hughes et al. [115] completed a systematic review and meta-analysis of HIV, hepatitis B and hepatitis C prevalence in people with SMI globally in 2016 [70]. The review found a higher prevalence of these viral infections in people with SMI in countries with a low prevalence in the general population and similar prevalence in countries with a high prevalence in the general population. The Hughes et al. [115] systematic review included many studies that were also included in the current review. The authors identified high levels of bias present in studies focusing on HIV prevalence in LMIC. They also commented on a general theme of higher HIV prevalence in women compared to men in both high and low prevalence countries, which is in keeping with findings from this scoping review. A 2016 systematic review [116] found that 10% of women with SMI experienced sexual violence, and that this was significantly higher than men with SMI and the general population. That review mainly included studies from HIC but also had similar findings to this review.

## What this review adds

Throughout the results, the burden of sexual and reproductive health needs for women with SMI in LMIC is clearly shown. Rates of STIs, HIV and risky sexual behaviour were high. A small number of studies on educational programmes for HIV showed positive outcomes. The need for more education on STIs and unwanted pregnancy prevention was identified. Studies based around pregnancy were often focused on the use of medication in pregnancy, but a small amount of research on pregnancy did state advantages (such as providing a purpose in life) and disadvantages (exacerbation of mental illness) of pregnancy for women with SMI.

The link between sexual dysfunction and SMI was found to be complex with multiple factors such as psychotropic medication, self-esteem and positive and negative symptoms of SMI to be involved. The overwhelming majority of studies about sexual violence found that women with SMI were at risk of sexual violence, including whilst admitted onto mixed inpatient wards, in the home environment or when homeless.

## Limitations

Limitations were present in this scoping review. Many LMIC were not covered in this review, and with the wide variation in contexts, cultures and geography, it is difficult to generalise these findings to all LMIC. Most studies were of a cross-sectional design, therefore causality cannot be implied for a majority of papers. As this is a scoping review, bias was not assessed. There may be high levels of bias in some studies, which could skew the conclusions of this review. Many studies had small sample sizes and are considered alongside larger studies, although an effort was made to note studies that were particularly small. Grey literature was excluded in this review due to the large amount of peer reviewed literature available. Capturing unpublished literature may ensure inclusion of more LMICs.

## Strengths

The strengths of this scoping review include a rigorous methodology with an iterative approach that resulted in early decisions to refine the inclusion criteria. Initially, only studies

that provided disaggregated data by gender and diagnosis were included. It was felt that some useful data was being excluded based on these criteria and therefore, the inclusion criteria was refined to include aggregated data where at least 50% of participants had SMI or 80% were women. A broad definition of sexual and reproductive health needs ensured that multiple areas of the review topic were included and ensured a broad overview of this multifaced area. Double coding was utilised throughout the review process to increase robustness of findings.

## Implications

### Implications for a subsequent review

This scoping review systematically identified literature on the sexual and reproductive health needs of women in LMIC in relation to study type, focus of study, population, setting and outcome. The nature of scoping review methodology is limited to descriptive or simple content analysis. To move to more interpretative thematic analysis, subsequent synthesis of evidence from one or more of the focus areas from this review is planned. To progress this, a mixed methods approach to evidence synthesis would enable interrogation and interpretation of data gathered from both qualitative and quantitative studies and facilitate better understanding of this complex area of health for women living with SMI in LMIC. This would enable the literature identified to be utilised to inform decision-making in policy and practice.

This review highlighted that only a small proportion of LMIC countries are represented and a future review will likely need to consider how applicable findings are for the culture and diversity of other LMIC countries and consider the use of grey literature.

### Implications for research and practice

There is a need for future research to better represent the breadth of LMIC countries and for high quality research to be carried out. Other key gaps in the literature were identified by this review. This included a need for greater focus on inclusion of people with lived experience as part of studies looking at sexual violence, as well as ensuring that ethical and consensual aspects are fully considered and reported by researchers. These are both areas underscored by the recent work of 'The Lancet Commission on ending Stigma and Discrimination in Mental Health', which outlined the increased effectiveness of interventions developed in co-production with people who have lived experience [117]. The Lancet Psychiatry Journal has also recently specified that research submitted for publication will be required to provide information on how people with lived experience have shaped research priorities, the design of the study and interpretation and write up of findings with authors expected to acknowledge this as a limitation of their work where it has not been carried out [118].

Marginalisation and social exclusion not only exacerbate access to mental and physical health care but are often the catalyst that leads to basic human rights of people with mental health conditions being contravened [117]. Future research would benefit from focussing on the effectiveness of interventions that integrate physical and mental health, most notably those aimed at reducing stigma and discrimination around mental health and sexual health needs, as well as culturally appropriate adaptations [117].

## Supporting information

**S1 File. Example search strategy.**
(DOCX)

**S2 File. Supplementary tables.**
(DOCX)

**S3 File. PRISMA checklist.**
(DOCX)

## Author Contributions

**Conceptualization:** Shilpa Sisodia, Charlotte Hanlon, Laura Asher.

**Data curation:** Zara Hammond.

**Formal analysis:** Shilpa Sisodia, Zara Hammond.

**Investigation:** Laura Asher.

**Methodology:** Jo Leonardi-Bee, Charlotte Hanlon, Laura Asher.

**Writing – original draft:** Shilpa Sisodia, Zara Hammond.

**Writing – review & editing:** Zara Hammond, Jo Leonardi-Bee, Charlotte Hanlon, Laura Asher.

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
