## [Decision Letter · Decision Letter 0]

11 Jul 2024

PONE-D-24-17681Sexual and Reproductive Health Needs of Women with Severe Mental Illness in Low- and Middle-Income Countries: A Scoping ReviewPLOS ONE

Dear Dr. Asher,

Thank you for submitting your manuscript to PLOS ONE. After careful consideration, we feel that it has merit but does not fully meet PLOS ONE’s publication criteria as it currently stands. Therefore, we invite you to submit a revised version of the manuscript that addresses the points raised during the review process.

The Reviewer found the manuscript to be a literature review on an important and timely topic that fills a gap in research in this area. The Reviewer also provided important suggestions to improve the manuscript and make it suitable for publication. I recommend that the authors proceed with a revision of the manuscript, taking care to address all of the reviewer's comments.

We look forward to receiving your revised manuscript.

Kind regards,

Stefano Federici, Ph.D.

Academic Editor

PLOS ONE

Additional Editor Comments:

The reviewer found the manuscript to be a literature review on an important and timely topic that fills a gap in research in this area. The reviewer provided important suggestions to improve the manuscript and make it suitable for publication. I recommend that the authors proceed with a revision of the manuscript, taking care to address all of the reviewer's comments.

Reviewers' comments:

Reviewer's Responses to Questions

**Comments to the Author**

1. Is the manuscript technically sound, and do the data support the conclusions?

Reviewer #1: Yes

2. Has the statistical analysis been performed appropriately and rigorously? 

Reviewer #1: N/A

3. Have the authors made all data underlying the findings in their manuscript fully available?

Reviewer #1: Yes

4. Is the manuscript presented in an intelligible fashion and written in standard English?

Reviewer #1: Yes

5. Review Comments to the Author

Reviewer #1: Thank you for allowing me to review the paper, “Sexual and Reproductive Health Needs of Women with Severe Mental Illness in Low and Middle-Income Countries: A Scoping Review”. This is an important and timely review which clearly lays out the existing research in this area and the gaps. I found this review very well written and the methods and results clearly articulated. I only have a few comments/suggestions which are outlined in the attached document.

6. PLOS authors have the option to publish the peer review history of their article (what does this mean?). If published, this will include your full peer review and any attached files.

Reviewer #1: No

---

## [Author Response · Author response to Decision Letter 0]

17 Sep 2024

We are grateful for the reviewers’ comments and have detailed our amendments and responses below. 

The following amendments are evidenced in the document titled: SRHSMILMIC.ScopingReview.FINAL. RevisedTrackChanges

Accepted changes are included in the document titled: 

SRHSMILMIC.ScopingReview.FINAL.Submitted.Manuscript

Background

1. It would be helpful to include a definition of what is meant by SMI within the background section. 

Response: Thank you for this comment. We have added the sentence ‘Severe mental illness (SMI) refers to mental health conditions associated with substantial and enduring impacts on functioning. In this review, our focus is on psychotic disorders such as such as schizophrenia, bipolar disorder and severe psychotic depression’ (Line 90-92)

Methods: 

1. The inclusion criteria is clear. However, it may also be helpful to include specific exclusion criteria applied.

Response: Thank you for this comment. We have added the following sentence detailing exclusion criteria ‘Studies that focused solely on child exploitation or on wider topics such as marriageability, marital relationships and gynaecological health were also excluded.’ (Line 137-139)

2. I can understand a potential rationale for not having a restriction on date of publication, but it would be helpful to outline why no restriction on date was chosen. 

Response: Thank you for this comment. We have updated the wording to reflect the rationale for not applying a restriction on date. This now reads ‘There was no restriction on study date or language to ensure the review captured any changes in literature publication over time.’ (Line 139-140)

3. It states there was no restriction on language. How were papers not published in English screened? 

Response: Thank you for this comment. We have now updated the wording to ‘It was decided that any titles, abstracts or full texts not available in English would be translated into English using online (DeepL or Google Translate) or native speaker translation services to allow screening.’ (Line 140-142)

‘All studies identified for screening were available with title and abstract in English language or French.’ (Line 147-148)

4. It may be helpful to reference the full example of a search that is currently in the supporting information in the text. Additionally, it would be good to reference that the checklist is also in the supporting information. 

Response: Thank you for this comment. This reference has now been added with he following amendment ‘A full example of the search strategy used for the review is available in Appendix 1.’ (Line 142-143)

5. I found Figure 3 confusing and difficult to interpret. For example, the difference in number of studies between Egypt and Peru is difficult to see. Is there another way to present this data? Maybe by including the n in the circles? You could even just use a bar chart showing frequency of publications for each country?

Response: Thank you for this comment. Figure 3 has been replaced to provide a tree map visual of countries included in the review and the n for each country. Title adjusted accordingly (Line 192 - 195)

Results

1. For the findings on the lack of reporting of ethical procedure, could this be explained by changes in norms of reporting ethical procedures throughout the years? For example, is this information mostly missing from papers published from over 10 years ago? If not, it is still interesting to present this data to show that even in recent publications this information is missing. 

Response: Thank you for this suggestion. Further analysis has been undertaken. No patterns/themes have been detected by publication year but info and analysis on country of publication is now included. 

‘Analysis by country of publication indicated that the greatest number of studies not reporting ethics approval were published in India (n=10) [24][38][35][36][37][3][44][52][66][67] but Tunisia had the greatest proportion with three [26][56][82] of the four studies not reporting this. Of the eleven studies that did not report all three areas of ethics approval, informed consent and capacity to consent, n=4 were published in India [24][44][52][67], n=3 were published in Tunisia [26][56][82], n=2 were published in Brazil[72][91] and n=1 were each published in Nigeria[105] and Turkey[45]. Analysis did not detect any patterns by year of publication with regards to ethics approval, informed consent and capacity to consent assessment among the included studies. (Line 248-255)

2. How were the six themes identified? I have seen more formal methods liked thematic synthesis or thematic analysis used to help identify themes for scoping reviews? Was a specific method used? Please explain what method/process was used and why? For example, were these identified themes agreed upon by the whole team? 

Response: Thank you for this feedback. The following sentence has been added for clarification ‘Quantitative data were summarised and topic areas within each study were coded. Once coded by topic, studies were grouped by frequency of topic and categorised accordingly.’ (Line 155– 156)

The abstract wording has been modified in line with the above and now reads ‘were grouped by frequency of topic into categories of…’ (Line 37-38)

3. In discussion of the themes, you could include some specific examples/more details from some of the included studies, especially those that include qualitative data to get a better sense of how the papers are discussing the identified needs in their specific contexts. Now the themes are very descriptive which help answer your first research question around extent and type of evidence. I would hope that some of the themes could be developed further to be more interpretive to help understand/summarise the needs identified (the last part of your research aim). 

For example, Pg 25 lines 364-365, you give some information about a specific study, but I would want to know a little more here. For example, what were the main perceptions and experiences of mental health care staff? As is done with thematic synthesis, you could even pull out some specific quotes from the paper. 

Response: Thank you for this feedback. We have not been able to tally the reviewer comments for page 25 with the content for that section. We feel a more interpretative approach and/or thematic synthesis would be beyond the scope of this scoping review. We have outlined this limitation and plans for subsequent reviews by editing and addition of the following wording ‘The nature of scoping review methodology is limited to descriptive or simple content analysis. To move to more interpretative thematic analysis, subsequent synthesis of evidence from one or more of the focus areas from this review is planned. To progress this, a mixed methods approach to evidence synthesis would enable interrogation and interpretation of data gathered from both qualitative and quantitative studies and facilitate better understanding of this complex area of health for women living with SMI in LMIC’. (Line 535 – 542).

In line with reviewer feedback, additional wording has been added and updated as follows:

Contraception & Family Planning: ‘In one study from Turkey, 73.7% of nurses stated that patients should receive family planning education however only 23.5% stated giving family planning advice to patients [34]‘ (Line 384 - 386). 

HIV (prevalence, risk-taking behaviour and knowledge): ‘One qualitative study from South Africa [41] identified barriers and challenges in delivering effective HIV counselling. Examples of how the health, social and wellbeing needs of women could be better met included improved data collection on HIV prevalence among psychiatric inpatients, capacity to support health literacy and ensure test results and diagnosis are fully understood and a need to improve follow up in the community after hospital discharge, particularly where a patient returns to a rural area [41].’ (Line 288 – 293) 

STIs (not including HIV): ‘One qualitative study drawing on interviews with mental healthcare providers in Brazil indicated that SMI patients’ decision making in relation to risk and sexual behaviour was impaired by their mental health illness, which led to unprotected sex and in turn a higher risk of STI than the general population [91]. Dialogue with patient focus groups echoed this, indicating there may be greater sexual health needs for this population in terms of risk perception, decision making and acquisition of STIs [91].’ (Line 308-315)

Discussion 

1. I wonder if the information in “Comparison to Previous Literature” could be moved to the background section? 

Response: Thank you for this suggestion. Our preference is to keep the comparison to previous literature in the discussion as this enables the findings of the review to be placed in the context of the previous literature. 

2. Pg 31, “Study type will need careful consideration as many of the studies included in 508 this review were observational and may provide rich insights but be challenging to draw conclusions from”. I think now there are some mixed-method systematic review methodologies that would help address this so I don’t think it is true that it would be challenging to draw any conclusions from these types of studies. 

Response: Thank you for this feedback. Wording has been updated to reflect this as follows ‘The nature of scoping review methodology is limited to descriptive or simple content analysis. To move to more interpretative thematic analysis, subsequent synthesis of evidence from one or more of the focus areas from this review is planned. To progress this, a mixed methods approach to evidence synthesis would enable interrogation and interpretation of data gathered from both qualitative and quantitative studies and facilitate better understanding of this complex area of health for women living with SMI in LMIC. (Line 535 - 542).

The following has been removed to ensure the editing additions of the above are consistent in the wording ‘There are challenges highlighted by this scoping review that will need to be addressed to progress to full evidence synthesis. Study type will need careful consideration as many of the studies included in this review were observational and may provide rich insights but be challenging to draw conclusions from.’ (Line 544 - 547)

---

## [Editor Report · Decision Letter 1]

23 Sep 2024

Sexual and Reproductive Health Needs of Women with Severe Mental Illness in Low- and Middle-Income Countries: A Scoping Review

PONE-D-24-17681R1

Dear Dr. Asher,

We’re pleased to inform you that your manuscript has been judged scientifically suitable for publication and will be formally accepted for publication once it meets all outstanding technical requirements.

Kind regards,

Stefano Federici, Ph.D.

Academic Editor

PLOS ONE

Additional Editor Comments (optional):

Although Reviewer #1 was not available to review the revised version, I believe that the authors have addressed all of Reviewer #1's requirements and that the manuscript is therefore suitable for publication.
---

## [Editor Report · Acceptance letter]

8 Nov 2024

PONE-D-24-17681R1 

PLOS ONE

Dear Dr. Asher, 

I'm pleased to inform you that your manuscript has been deemed suitable for publication in PLOS ONE. Congratulations! Your manuscript is now being handed over to our production team.

Kind regards, 

on behalf of

Prof. Stefano Federici 

Academic Editor

PLOS ONE